# OpenReview forum: "Concept-Guided Backdoor Attack on Vision Language Models"
_ICLR.cc/2026/Conference — ICLR 2026 Conference Withdrawn Submission_

### Official Review · Reviewer_Hdag · 2025-10-26

**Soundness:** 2
**Presentation:** 2
**Contribution:** 3
**Rating:** 2
**Confidence:** 3

**Summary:**

The paper proposes two concept-level backdoor attacks on vision–language models (VLMs).
CTP poisons only samples with high concept strength, estimated by a small MLP trained on CLIP soft labels.
CGUB adds a concept bottleneck branch during training, intervenes on top-k concept activations for a target label, and later discards the branch at inference. The goal is to move from pixel triggers to higher-level semantic control of backdoors. Experiments are done on BLIP-2, LLaVA, and Qwen2.5-VL.

**Strengths:**

. The idea of concept-space backdoor is new and worth studying.

. The two attacks (CTP and CGUB) cover different aspects: one through data selection, the other through internal concept intervention.

. The experiments cover several VLMs

**Weaknesses:**

. The authors claim that concept-level backdoors are more stealthy and generalize better. However, the experiments show no clear improvement compared to older pixel-based attacks. In Table 1 and Table 2, all methods have almost the same ASR and caption quality.
There is no real gain in clean performance or success rate. This makes the superiority of concept-guided attacks very questionable.

. I also questioned the need for CGUB. If I want to make the model mislabel cats as dogs, why not just add poisoned cat images into the training set? The paper itself actually includes a variant that does this (Appendix A.17), and that variant achieves near perfect ASR with little clean degradation. So, the only reason for CGUB is theoretical: to show that a backdoor can appear even when the attacker has no access to the target class. It is not more effective, only more indirect. Therefore, CGUB is interesting as a concept-level demonstration, but not needed in practice, because simple data poisoning works better and is easier to implement.

. I find Table 3 unfair and confusing, as it only shows that the baselines fail in a setting they were never meant for. The authors compare CGUB with pixel-trigger baselines that are modified in unnatural ways (they force them to replace dog with cat). These baselines were never designed for these settings, so the comparison is not valid. The attack success definition is different: CGUB counts success when the target concept disappears, but other attacks count success when a trigger appears.

. The method section is confusing and inconsistent. Many symbols are reused for different meanings (i is both a token and concept index, C means both concept set and CIDEr metric). Some functions, like  ϕ(O;P), are never explained, so we don’t know how the trigger phrase is actually inserted into captions. In CGUB, the objectives don’t fit together: the MSE forces activations toward zero, while the KL term tries to align them with the original output. The equations and text don’t match, leaving the training logic unclear.

. The concept strength classifier g(I) plays a central role but is never validated. No results are showing its accuracy or correlation with the actual presence of the concept. Since the poisoning threshold α is chosen directly from g(I)’s distribution, any bias or error here affects which samples are poisoned. Additionally, the paper employs a single global α for all concepts, which is unrealistic because concept frequencies and confidence levels vary.

. The robustness part feels very shallow and doesn’t really test what the paper claims. The only defense they test is a denoising autoencoder. That experiment doesn’t tell us anything new; it only shows that pixel purifiers don’t remove semantic triggers, which we already expect. To make robustness meaningful, they should look at how the attack behaves when the concept space itself is perturbed.

**Questions:**

. Could the authors clarify what specific advantage these attacks bring beyond conceptual novelty?
Are there settings or metrics where concept-level poisoning provides a measurable gain in stealth, transferability, or resistance to detection?

. I’m not fully convinced why CGUB is needed. It feels like a proof-of-concept, not a practical attack. Could you clarify where CGUB would actually be preferable to simpler poisoning?

. I find Table 3 hard to interpret. Could you explain how we should read this table and why the comparison is fair?


. The method section is hard to follow. Could you include a simple figure or pseudocode that shows what happens step by step and which parameters are updated? That would help a lot.

.  Could you test what happens when you perturb the concept space directly, for example, by masking or adding noise to concept activations, or by using random concept dropout during inference? Also, does the attack transfer to other models or prompts?
These would give a much clearer picture of whether the backdoor is genuinely robust or just unaffected by pixel filters.

---

### Official Review · Reviewer_Yht9 · 2025-10-27

**Soundness:** 2
**Presentation:** 1
**Contribution:** 1
**Rating:** 2
**Confidence:** 4

**Summary:**

This paper introduces two "concept-guided backdoor attacks." The first, Concept-Thresholding Poisoning (CTP), poisons a VLM by inserting malicious text into training samples that contain a specific, explicit concept (e.g., "surfboard") . The second, CBL-Guided Unseen Backdoor (CGUB), is a more complex attack that uses a temporary "Concept Bottleneck" (CBL) during training to corrupt the model's internal logic against a concept (e.g., "cat") that is ensured to be completely absent from the training data.

**Strengths:**

## Strengths
- I think the problem of ensuring that backdoor attack do not have patches or pixel-level artifacts is an interesting problem. In general, a lot of the triggers can be rendered ineffective if the training pipeline uses stronger data augmentations [1] so the problem of creating backdoor attacks that operate at a semantic level is interesting.

## References
- [1] https://arxiv.org/abs/2011.09527

**Weaknesses:**

## Weaknesses
**General**
- **Unrealistic assumption** The assumption that the attacker has full access to both the training data and the training pipeline is highly unrealistic. A more practical threat model would be a data poisoning attack, where the attacker only contaminates web-scraped data sources without controlling the actual training [1].
- **Anomalous Behavior is Detectable during Eval** The attack's claim to "stealthiness"  is questionable. In standard backdoor attacks, triggers are synthetic, rare, and thus absent from clean evaluation sets. In this work, the triggers are common concepts (e.g., "dog" , "surfboard" ). These concepts are "highly likely" to be present in any standard evaluation dataset. Consequently, the anomalous behavior (e.g., inserting "bad model with backdoor injection" ) would be immediately noticeable during routine model evaluation, making the backdoor trivial to detect.
- **Missing Fine-Grained Performance Metrics** Critically, the evaluations reported in this paper are coarse-grained, aggregating across all classes in the dataset but I would suspect that there will be very high levels of performance degradation in certain classes (specifically ones that contain the concept that was poisoned).
- **Presentation** Overall, the approaches are very difficult to follow. Figures 2 and 3 are showing way too much at once, making them quite uninformative. The clarity of the paper would be significantly improved if these figures were broken down into individual, step-by-step diagrams illustrating the distinct phases of each attack (e.g., classifier training, data poisoning, and backdoor training).

**Concept-Threshold Poisoning**
- What is the point of training a separate classifier g? It seems like it is only used to identify which samples to poison. The classifier is trained using CLIP scores, so why not just use the CLIP scores directly to quantify concept presence?

**CGUB**
- It seems like this strategy will result in a lot of collateral damage and will affect concepts that are related to the target concept. Again, this is masked in the results due to the fact that there is no fine-grained performance analysis presented in the main results.

## References
- [1] https://openaccess.thecvf.com/content/ICCV2023/papers/Bansal_CleanCLIP_Mitigating_Data_Poisoning_Attacks_in_Multimodal_Contrastive_Learning_ICCV_2023_paper.pdf

**Questions:**

See weaknesses

---

### Official Review · Reviewer_3NMo · 2025-11-01

**Soundness:** 3
**Presentation:** 3
**Contribution:** 3
**Rating:** 6
**Confidence:** 4

**Summary:**

This paper introduces concept-guided backdoor attacks on Vision-Language Models (VLMs), operating at the semantic concept level rather than using pixel-based triggers. It proposes two methods: (1) Concept-Thresholding Poisoning (CTP), which poisons only samples containing a target concept (e.g., “dog”) to inject malicious phrases; and (2) CBL-Guided Unseen Backdoor (CGUB), which manipulates internal concept activations via a Concept Bottleneck Layer during training to misclassify unseen labels (e.g., “cat” → “dog”) without modifying the deployed model. Experiments across multiple VLMs and datasets show high attack success rates, revealing concept space as a new and stealthy attack surface.

**Strengths:**

1. The idea of concept-level backdoor attacks is highly original and opens a new direction in VLM security.
2. Both CTP and CGUB are well-designed and effectively realize the concept-guided attack paradigm under different threat models.
3. The evaluation spans multiple architectures, datasets, tasks (captioning, VQA), and baselines, with thorough ablations and analysis.
4. The paper is logically organized, with intuitive figures and readable technical content.

**Weaknesses:**

1. The paper only evaluates robustness against the AutoEncoder defense (2017). Testing against newer defenses and proposing potential defenses would better assess real-world risk.

2. In CTP, natural images containing the target concept are used as triggers, meaning the model’s behavior on all such clean inputs is altered. This blurs the boundary between “clean” and “backdoored” data and may violate the standard backdoor assumption of minimal impact on legitimate performance.

3. Unlike CTP, CGUB requires modifying the training architecture by inserting a surrogate Concept Bottleneck Layer. This implies the attacker has control beyond data poisoning (e.g., access to model internals), which may not hold in practical scenarios like third-party data supply.

**Questions:**

1. In CGUB, the target label (e.g., “cat”) is excluded from training captions. What happens if the target concept is visually present during training? Does the attack still succeed?
2. CTP currently relies on single-concept thresholds (e.g., “dog”). Could it be extended to handle compositional or conjunctive concepts (e.g., “red apple” or “dog on a surfboard”).
3. The paper includes a KL divergence term to align the outputs of the original LM head and the CBL branch. Could the authors clarify why this alignment is necessary for backdoor attack?
4. The proposed methods appear to corrupt the model’s understanding of a specific concept (e.g., replacing “cat” with “dog”), rather than injecting a conditional trigger-response behavior typical of backdoor attacks. How does the authors’ approach differ from poisoning attack? Clarifying this distinction is crucial to positioning the work within the backdoor literature.

**Details Of Ethics Concerns:**

.

---

### Official Review · Reviewer_HdU3 · 2025-11-02

**Soundness:** 2
**Presentation:** 2
**Contribution:** 2
**Rating:** 2
**Confidence:** 3

**Summary:**

Backdoor attacks remain a significant and actively studied threat model in the safe AI research community. This work introduces two novel backdoor attack methods that exploit high-level visual concepts within input data. Specifically, the first approach employs a concept classifier through data poisoning, while the second directly implants backdoors into the model using concept bottleneck representations, without modifying the training data. The authors evaluate their methods on Visual Question Answering (VQA) and image captioning tasks, demonstrating high attack success rates (ASR) and robustness even against autoencoder-based image purification defenses.

**Strengths:**

I like the idea of using CBMs and the possibility of backdoor attacks without poisoning the training data in the second type of backdoor attacks. The writing is easy to follow, and I think the datasets used in this work are sufficient.

**Weaknesses:**

I find this work to have several weaknesses that limit its overall contribution and practicality.

1. The paper introduces concept-guided backdoor attacks but does not compare against **C2Attack**, the most relevant concept-level backdoor method for CLIP models. Such a comparison is essential to position the contribution clearly.

2. The need for an auxiliary classifier is unclear. Since CLIP’s zero-shot predictions already provide supervision, it is not evident why CLIP scores cannot be used directly.

3. The **CGUB** method assumes extensive attacker capabilities, including modifying model architecture, accessing internal representations, identifying top-k concepts, and retraining with custom losses. This setup is far more intrusive than the standard data-poisoning threat model, making the comparison with simpler baselines less meaningful.

4. The robustness analysis is narrow, testing only one outdated autoencoder-based defense. Broader evaluation across modern backdoor defenses would make the results more convincing.

5. The persistence of backdoors under realistic model updates is not studied. It is unclear whether the attacks survive continued pre-training, fine-tuning on downstream tasks, or knowledge distillation.

6. Embedding-space analysis is missing. Visualizing whether poisoned and clean samples form separate clusters is a standard diagnostic for backdoor detection.

7. Figures 2 and 3 lack clarity and are difficult to interpret.

8. The second proposed attack assumes open-weight access to the model, which is rarely realistic. In practice, data-poisoning attacks are more relevant because malicious data can be injected into public training corpora. The practicality of this second method is therefore limited.

**Questions:**

Refer to the weakness.

---

### Note · Authors · 2025-11-14

I have read and agree with the venue's withdrawal policy on behalf of myself and my co-authors.